# Authenticity and Typicity of Traditional Cheeses: A Review on Geographical Origin Authentication Methods

**DOI:** 10.3390/foods11213379

**Published:** 2022-10-26

**Authors:** Marco Cardin, Barbara Cardazzo, Jérôme Mounier, Enrico Novelli, Monika Coton, Emmanuel Coton

**Affiliations:** 1Department of Comparative Biomedicine and Food Science, University of Padova, Viale Università 16, 35020 Legnaro, PD, Italy; 2Univ Brest, INRAE, Laboratoire Universitaire de Biodiversité et Écologie Microbienne, F-29280 Plouzané, France

**Keywords:** cheese, geographical origin, authentication, next-generation sequencing, volatilome, isotopic analysis, trace element analysis, infrared fingerprinting

## Abstract

Food fraud, corresponding to any intentional action to deceive purchasers and gain an undue economical advantage, is estimated to result in a 10 to 65 billion US dollars/year economical cost worldwide. Dairy products, such as cheese, in particular cheeses with protected land- and tradition-related labels, have been listed as among the most impacted as consumers are ready to pay a premium price for traditional and typical products. In this context, efficient food authentication methods are needed to counteract current and emerging frauds. This review reports the available authentication methods, either chemical, physical, or DNA-based methods, currently used for origin authentication, highlighting their principle, reported application to cheese geographical origin authentication, performance, and respective advantages and limits. Isotope and elemental fingerprinting showed consistent accuracy in origin authentication. Other chemical and physical methods, such as near-infrared spectroscopy and nuclear magnetic resonance, require more studies and larger sampling to assess their discriminative power. Emerging DNA-based methods, such as metabarcoding, showed good potential for origin authentication. However, metagenomics, providing a more in-depth view of the cheese microbiota (up to the strain level), but also the combination of methods relying on different targets, can be of interest for this field.

## 1. Introduction

The shared definition of food fraud relates to intentional illegal acts performed by food value chain operators for economic gain [1]. More specifically, in the framework of the European agri-food chain legislation, food fraud is defined as “any suspected intentional action by businesses or individuals for the purpose of deceiving purchasers and gaining undue advantage therefrom, in violation of the rules referred to in Article 1(2) of Regulation (EU) 2017/625” [2]. Behind the term “food fraud”, multiple practices designed to deceive purchasers, which are categorized under the following denominations: (i) substitution, (ii) concealment, (iii) dilution, (iv) unapproved enhancement, (v) counterfeit, (vi) grey market/forgery, and finally, (vii) mislabeling, exist. Substitution corresponds to total or partial replacement of a food, including ingredients or nutrients, with one of lower value. Concealment hides the low quality of food ingredients or food products. Dilution is self-explanatory and corresponds to the action of mixing a high-value ingredient with a lower one, while unapproved enhancement improves food quality by adding undeclared or unknown ingredients. These four food fraud types are grouped under the term “adulteration”. Counterfeit refers to the infringement of Intellectual Property Rights via replication of a product or its packaging, while grey market or forgery corresponds to production, theft, and diversion involving unauthorized sales of foodstuffs. The latter generally concerns products for which production agreements or quotas exist or geographical restrictions apply. Finally, mislabeling corresponds to distorted information or false claims on packaging or labels [3].

The global estimated value of food frauds each year ranges from 10 to 65 billion US dollars, without considering potentially related losses [1]. Indeed, unfair competition may not only result in economic losses for honest producers and retailers but could also impact food safety and quality, public health, and society at a large scale, thus impeding a perfectly accurate estimation of food fraud socio-economic impacts [4,5].

In recent years, dairy products have been systematically listed among the most common food frauds [6,7,8,9,10,11], with cheese being the most prevalent [11,12]. In this case, fraudulent documentation and adulteration/substitution were the most frequent events [12,13]. Between 2000 and 2018, the HorizonScan program (a subscription-based service monitoring global food integrity issues, including brand identity) reported 245 cases of dairy frauds, from which 51% were characterized by fraudulent documentation [12]. Similar findings were reported by Montgomery et al. [13], who stated that cheese fraudulent documentation accounted for 74% of total fraud cases (*n* = 98) between 2015 and 2019 [10,11]. In this context, actions against food fraud are taken by inspecting agencies, producers, and retailers [14,15,16,17]. Nevertheless, the increased complexity of a globalized supply chain can impair fraud incidents from being detected.

The term “authenticity” for food products is associated with the fact that there is a “match between the food product characteristics and the corresponding food product claims” [18]. In this context, cheeses are defined as ripened or unripened soft, semi-hard, hard, or extra-hard products, obtained by milk protein coagulation using rennet, other suitable coagulating agents, or processing technologies, and have whey protein/casein ratios that do not exceed that of milk [19]. Cheese quality is often linked to value descriptors such as environmental welfare standards, production methods, and safety claims, but also geographical origin [20].

As for the term “typicity”, it is defined by the unique combination of natural and human factors associated with a specific terroir [21]. Cheese typicity (i.e., the recognizable organoleptic traits associated with a given cheese) is acquired from the specific raw materials used, traditional tools, and the encountered environmental and production conditions, cheese-making process, and geographical area [22]. At the European level, linked to this typicity, certain cheeses can be recognized with distinctive labels, such as Protected Designation of Origin (PDO), which indicates that the products were entirely manufactured in a defined geographical area, Protected Geographical Indication (PGI), which correlates a geographical area with at least one of the product transformation steps, and Traditional Specialties Guaranteed (TGS), that highlights a traditional aspect for food products without any link to a specific geographical area. Other labels, such as “product of island farming” or “mountain product”, can also be found [23,24,25]. A strong societal demand currently exists for natural, local, traditional, and authentic foods. Authentic food quality is recognized by the consumer to have a higher added value, but this higher value increases the risk for fraud, in particular fraudulent documentation including omission or irregular use of geographical origin and failure to adopt suitable traceability systems (i.e., corresponding to mislabeling) [13,26]. At the European level, Regulation [27] established the implementation of a comprehensive traceability system within food businesses and required a suitable documentation system to identify the product along the food chain, while Regulation [28] (Art. 26) requires labeling “country of origin” for products such as Geographical Indication and meat or products, for which mislabeling would mislead the consumer. Moreover, the increasing consumer demand for natural, local, and traditional foods has led to national laws, such as in Italy and Spain, regulating how to label the geographical origin of milk and milk derivatives [29,30]. Accordingly, geographical origin, described as a “specific location” that serves to designate a product origin such as territory of a member, region, or locality in a given territory [31] (Art. 22), has crucial relevance in dairy products. The need for food authentication methods is driven by different actors. This includes producers and retailers for whom food fraud induces economical losses, public authorities that verify compliance with agri-food chain legislation, and finally, the consumer, to ensure trust when buying a product. While, in the past, authentication analyses focused on evaluating a single molecule or single parameters, nowadays, these methods are evolving from targeted to untargeted approaches. This enables the description of multiple product features and characteristics to provide a way to develop fingerprints for cheese geographical origin authentication. This is particularly relevant for protected land- and tradition-related labeled cheeses that are often incriminated in food frauds due to their high economical value.

Two different strategies can be employed to authenticate cheese origin. The first one involves exploring the relationship between the biological and/or chemical components, assuming that their proportions are constant for a particular cheese at a specific time during production or shelf-life. In this context, it seems clear that pattern recognition methods (e.g., such as principal component analysis (PCA) and orthogonal projections to latent structures discriminant analysis (OPLS-DA)) create unique classes, potentially differentiating typical and fraudulent products [32]. The second strategy aims to find specific chemical or biological components which can be used as markers for traditional cheese authentication (e.g., mass spectrometry analysis, such as stable isotope ratio and trace elements) that can all reflect cheese chemical composition. DNA-based methods for authentication are also emerging in the dairy sector, as shown recently by the work of Kamilari et al. [33], who used cheese microbiota metabarcoding for this purpose. However, for authentication of geographical origin, methods such as isotopic profiles [34] are generally preferred. As food labeling systems are constantly evolving, in parallel with legislation (e.g., Regulation [2]), analytical tests and technical control measures need to be improved and updated to counteract present and emerging fraud systems [35].

In this context, this review (based on 167 articles published over the last 27 years) aims to present methods, either physical and chemical or DNA-based, that are currently used for cheese geographical origin authentication, highlighting their principle, application, discriminative power, and advantages and limits, as well as to present future perspectives in this analytical field.

## 2. Chemical and Physical Methods for Cheese Origin Authentication

Polyphasic chemical and physical analysis approaches are now more common than single-parameter descriptions (i.e., dry matter, total protein, salt content) [36] to decipher cheese composition profiles. This is reinforced by the fact that cheese characterization, based on general chemical parameters, does not efficiently discriminate cheese geographical origin, as recently shown for water buffalo mozzarella by Salzano and colleagues [37]. That is why multiple signals are analyzed to acquire specific insight into typical cheese characteristics connected with its origin. Isotope and trace element fingerprinting methods have been considered as reference methods; however, other chemical and physical analyses can be used for geographical origin authentication.

### 2.1. Stable Isotope Ratio Mass Spectrometry

Stable isotope ratio mass spectrometry (IRMS) is among the most common methods for food geographical origin authentication [3]. It detects natural isotopic abundance of light and heavy stable isotopes, which mostly depend on climatic or geographical conditions (mainly latitude and altitude). The stable isotope ratio is also affected by biological and environmental interactions, and thus geographical product origins can be differentiated even if these have a high degree of similarity. Elements are called isotopes when their atoms are made by the same number of protons but a different number of neutrons, yielding a different atomic mass than the normal element [38]. Stable isotopes are non-radioactive isotopes and do not decay rapidly to form other elements.

Usually, stable isotope analysis is expressed as a ratio using an international standard to calculate it (Equation (1)):δ‰ = (R_Sample_ − R_Standard_)/R_Standard_ ∗ 1000(1)
where R is the ratio between heavy and light isotopes [39]. The results of stable isotope ratios are always expressed as a percentage (‰—per mille unit) of international standard samples received from international organizations such as Vienna-Pee Dee Belemnite (V-PDB) for δ^13^C, Aria (AIR) for δ^15^N, Vienna—Standard Mean Ocean Water (V-SMOW) for δ^2^H and δ^18^O, and Vienna—Canyon DiabloTriolite (V-CDT) for δ^34^S [40].

IRMS has been widely used for cheese origin authentication [41,42,43] and many stable and unstable isotope ratios have been investigated. Among them, ^13^C/^12^C, 15N/14N, ^2^H/H, ^18^O/^16^O, and ^34^S/^32^S elements are the most commonly used, while ^87^Sr/^86^Sr, ^44^Ca/^40^Ca, ^44^Ca/^42^Ca, and ^206^Pb/^204^Pb are occasionally reported [38,44]. For cheese authentication, IRMS is based on predictable and reproducible responses of stable isotopes to typical factors such as geographical origin, animal origin, seasonality, and manufacturing processes [38]. Animal feed was shown to have the highest impact on δ^13^C and δ^15^N, while δ^2^H is heavily influenced by the animal diet, and its combination with δ^18^O is mainly impacted by geographical origin and seasonality [38,41]. δ^34^S is mostly linked to geographical origin, i.e., soil geology, and it is not correlated with other stable isotope ratios [45]. On the contrary, further studies are still needed to gauge the effect of cheese-making on stable isotope ratios [38] as different results on these ratios in milk and cheese have been reported. They suggested either no major difference between milk and cheese obtained after milk processing [41,43] or a partial impact [34,42]. Considering isotope abundances in different organic macromolecules, the casein fraction has been reported to be the most reliable for origin authentication [39].

In general, possible effects of the cheese-making process on stable isotope ratios could be related to fat removal (in particular, the glycerol fraction), curd acidification, curd clotting (e.g., use of a commercial starter vs. natural milk cultures), curd washing, curd heat treatment (e.g., 50 °C), and salt washing/brining, while ripening time has not yet been reported to impact their composition [38,46]. For the casein stable isotope ratio of milk, the corresponding cheese did not show any significant differences for δ^13^C, δ^15^N, and δ^2^H, but an unexplained and significant difference (*p* < 0.001) was reported for δ^18^O in typical pressed-cooked cheese [41]. The authors suggested a relevant fractionation in the animal in comparison with the feed, but a lack of isotopic fractionation during cheese-making [41]. However, Bontempo et al. [34] obtained a different isotopic ratio comparing milk and corresponding Mozzarella di Bufala Campana PDO for δ^2^H and δ^18^O. Further studies on milk and corresponding cheeses obtained through different processes and technologies may provide additional insight into the eventual changes of stable isotope ratios during cheese-making. Table 1 reports the advantages and limitations of stable isotope ratio mass spectrometry for cheese origin authentication.

From an applied point of view, IRMS was used to discriminate the origin of two typical mountain cheeses from Italy [41]. The authors combined δ^13^C, δ^15^N, δ^2^H, and δ^18^O from the casein fraction to build a canonical discriminant model that, after cross-validation, was able to correctly classify 96% of the milk and cheese samples. In Brazil, Silva and colleagues [42] analyzed δ^13^C, δ^15^N, δ^2^H, and δ^18^O from the water fraction of milk and cheese. After using the δ^2^H and δ^18^O in a linear regression model, they were able to discriminate milk and cheese samples according to their production area. In another study, Pillonel and colleagues [48] showed that the combined measurement of δ^13^C, δ^15^N, δ^18^O, and δ ^34^S/^32^S of the casein fraction authenticated Swiss vs. French Raclette cheeses.

In summary, isotope fingerprinting has proven to be reliable for the geographical origin authentication of typical cheeses. It is worth noting that stable isotope ratio analyses are already used as a traceability tool for some PDO cheeses such as Grana Padano and Parmigiano Reggiano (Regulation (EU) No. 1151/2012, amendment 2017/C 358/10 and 2018/C 132/07). This method is also often combined with trace element determination using inductively coupled plasma-mass spectrometry (ICP-MS) or inductively coupled plasma-atomic emission spectroscopy (ICP-AES).

### 2.2. Inductively Coupled Plasma

Cheese elemental fingerprinting is also currently used for origin authentication using inductively coupled plasma methods (ICP-MS or ICP-AES). These analyses rely on electromagnetic plasma to induce atom ionization (MS) or excitation (AES), to detect the different elements. Four main steps, including sample introduction and aerosol generation, plasma ionization/excitation, signal discrimination, and detection are used in ICP-MS and ICP-AES [44]. While MS analyzes ionized elements, AES detects light emitted by excited atoms. For cheese geographical origin authentication, ICP-MS is the preferred choice since it is a rapid, multi-element analysis able to quantify trace (ppm–ppb) or ultra-trace (ppb–ppq) elemental concentrations [49]. As elemental composition is mainly affected by geological and pedological traits, multiple factors can affect the element content in cheese, such as animal breed, feed vegetation, drinking water, and mineral supplementation [49,50]. Possible effects of cheese-making on cheese elements could derive from the clotting agent, curd acidification, manufacturing equipment, curd washing, and salt washing/brining [34,51]. Indeed, some authors reported that exclusion of Cu^2+^ and Zn^2+^ in multivariate analysis was necessary since a high transfer rate of these elements from dairy equipment to cheese was expected [52]. Table 2 reports the advantages and limitations of inductively coupled plasma for cheese origin authentication.

Elemental fingerprinting data obtained by Danezis and colleagues [50] have provided useful insights for an understanding of multiple element variations, including rare earth and precious metals, occurring in Greek Graviera cheeses obtained from 9 different regions (*n* = 105 samples). These authors analyzed 61 different elements, including rare earth (Dy, Er, Eu, Nd, Pr, Sc, Sm, Y, Yb), precious metals (Au, Pd, Re, Ru), ultra-trace elements (Nb, Ta, Tl, W, Zr), and trace elements (Ag, Al, Bi, Cd, Cu, Mo, Ni, Pb), and found significant differences according to cheese origin. They were able to correctly classify 92.1% of the tested cheeses (21 traditional and 9 commercial cheeses) using discriminant analysis [53]. The model was built on 65 elements, but the most significant variables were Ce, Dy, Eu, Gd, Ho, La, Nd, Pr, Sm, Tb, Y, Yb, Pd, As, Ba, Co, Fe, Ga, Mo, Ni, Ti, Zr, Ca, and P. Even if the authors did not perform any cross-validation, their results showed that ICP-MS was useful for cheese origin authentication, achieving high correct classification rates. According to the same study, rare earth elements seemed to significantly increase the discriminatory power of ICP-MS.

As previously stated, a combination of IRMS and ICP-based methods can also be used. For example, IRMS (for the casein fractions) and ICP-MS achieved high performance for cheese origin classifications. This was shown for semi-cooked typical Italian alpine cheeses, namely Asiago, Fontina, Toma, Vezzena Montasio, Spressa, and Puzzone (*n* = 109). The use of Ba, Ca, Cu, Ga, K, Mg, Rb, Zn, δ^13^C_casein_, δ^15^N_casein_, δ^13^C_glycerol_, and δ^18^O_glycerol_ as predictive variables in canonical discriminative analysis was able to correctly classify 94% of the tested samples [52]. Another example was for authentication of PDO Parmigiano Reggiano vs. 11 imitation cheeses from different origins [54]. In their process, Camin et al. [54] used variables selected by the Random Forest algorithm (δ^13^C_casein_, δ^2^H_casein_, δ^15^N_casein_, δ^34^S_casein_, and Sr, Cu, Mo, Re, Na, U, Bi, Ni, Fe, Mn, Ga, Se, and Li) to create a supervised two-class model that was able to correctly classify 98.3% of the 265 hard cheese samples. Lastly, Nečemer and colleagues [51] discriminated different Slovenian cheese origins by combining P, S, K, Cl, Ca, Zn, and δ^13^C_casein_ and δ^15^N_casein_ contents, and a high correct classification rate (97%) confirmed that the dual IRMS/ICP-MS approach provided robust data to authenticate cheese geographical origins.

### 2.3. Infrared Spectroscopy

Spectroscopic analyses, including near-infrared (NIR) and mid-infrared (MIR), are based on the selective interaction of infrared beams with food molecules [55]. The infrared region includes a wide energy range (800 nm–1 mm, 12,500–10 cm^−1^) and spectrophotometers can only evaluate a fraction of the wavelength, such as NIR (800–2500 nm, 12,500–4000 cm^−1^), MIR (2.5–25 µm, 4000–400 cm^−1^), and far-infrared (25–1000 µm, 400–10 cm^−1^) [56]. Although infrared spectroscopy mainly involves vibrational energy, NIR incorporates both electronic and vibrational spectroscopy, while MIR mainly monitors molecular vibrations and far-infrared contains rotatory and vibrational movements. NIR spectroscopy mainly reflects the absorption information of overtone and combination tone of chemical bond vibrations of hydrogen-containing groups (C–H, O–H, N–H, and S–H) that reflect the anharmonic constant and the high-frequency vibration of the fundamental stretching of a XH bond (i.e., second overtone transition of C–H and O–H in the 1050–1400 nm region) [55,56]. Bands in the NIR region are weak or very weak, making this region markedly different from the others, but at the same time, more difficult to analyze. However, compared to the MIR region, OH and NH stretching bands of monomeric and polymeric species are better separated and differentiate free terminal functional groups from those within the molecule [56]. On the other hand, MIR spectroscopy is a highly sensitive method in which polar functional groups such as C=O, OH, and C–S exhibit intense bands. These bands, combined with other specificities of this region, such as stronger bands from antisymmetric vs. symmetric stretching, make this analysis useful for molecular fingerprinting [56].

In food fingerprints, spectroscopic techniques have gained popularity since these are fast, solvent-free, automatic, non-destructive, non-invasive, inexpensive, and can be used as a multiparameter analysis [49]. In general, NIR spectroscopy (in reflectance or transmittance mode) is more often used than MIR spectroscopy for food analysis as it requires less sample preparation and can be easily used for in-field analysis [57].

Cheese has proven to be a challenging matrix for infrared spectroscopic analysis as it is non-homogeneous (e.g., crystalline structure, holes) and numerous cheese types exist [58]. Nevertheless, many studies on cheese characterized and correctly predicted the chemical composition, manufacturing technique, ripening time, seasonality, and feeding system of milk-producing animals [59,60,61,62]. On the other hand, only a limited number of studies have focused on cheese origin authentication, possibly connected to the initial inability to differentiate milk geographical origin using infrared spectroscopic techniques [63] and sample size needed to validate the analysis [61]. Table 3 reports the advantages and limitations of infrared spectroscopy for cheese origin authentication.

According to Niermöller and Holroyd [64], studies conducted on a reduced sample size (*n* < 60) reported weak calibration for NIR spectra since, according to the multivariate model employed for the chemometrics analysis, sample size should be higher; in fact, PLS may require hundreds of samples per category. In another study, NIR spectroscopy correctly classified 96% of cheese samples from pasture-fed and conserved-forage-fed (hay or grass silage) dairy cattle [66]. More recently, NIR was used (850 to 1048 nm wavelength region) for a cheese model obtained from different dairy systems, and 67.1% of samples were correctly classified after applying a cross-validation (LDA model) [62]; however, the aim was not to discriminate the geographical origin of the tested samples. In a study comparing MIR and NIR spectroscopy performances, Karoui et al. [67] correctly classified 86.6% and 85.7% of 91 Emmental PDO cheeses obtained from Switzerland, France, Finland, Germany, and Austria using factorial discriminant analysis, thus highlighting MIR and NIR performances to discriminate cheeses produced with a similar process but with different geographic origins. In another study, Karoui et al. [68] used two MIR regions (3000–2800 and 1500–900 cm^−1^) to successfully authenticate PDO Gruyère and L’Etivaz cheese, and 90.5% and 90.9% of samples were correctly classified using a factorial discriminant analysis.

In conclusion, although some studies have tested NIR and MIR spectroscopy to authenticate cheese origins, it is still difficult to conclude to what extent these techniques can be applied given the relatively low number of samples used in these studies [61,65]. In this context, further work on larger sample sizes for diverse cheese categories will likely improve the discriminatory power of these methods.

### 2.4. Nuclear Magnetic Resonance

Another spectroscopic analysis based on selective interactions between electromagnetic radiation and sample molecules is nuclear magnetic resonance (NMR). Generally, NMR uses radio frequency pulses to induce magnetic resonance nuclei-oriented transitions in an external magnetic field. When electromagnetic radiation hits nuclei, oriented nuclei move from lower to higher energy status or resonant nuclei. Subsequently, nuclei emit energy to return to the lower energy status, producing free induction decay [69]. The detected energy produces an absorbance signal, expressed in ppm, obtained from the ratio of standard molecules (e.g., 3-(trimethylsilyl)-propionate-d4). NMR, based on nuclei magnetic angular momentum—spin—is characterized by the azimuthal quantum number (I). Only nuclei with an even number of neutrons and an odd number of protons can be detected by their magnetic angular momentum (e.g., ^12^C and ^16^O have I = 0, while ^1^H and ^13^C have I > 1). Different NMR methods have already been employed in food authentication. However, considering cheese geographical origin authentication, ^1^H high-resolution magic angle spinning (HRMAS) NMR is the most commonly used technique [70,71]. For example, Shintu and Caldarelli [71] applied a discriminant analysis using unsaturated fatty acid, aspartic acid, serine, and olefinic proton signals to classify the geographical origin of 20 Emmental cheeses from Austria, Finland, France, Germany, and Switzerland. They correctly classified 89.5% of samples after cross-validation. Mazzei and Piccolo [70] successfully identified Mozzarella di Bufala Campana PDO cheese origin. The studied cheeses were obtained in the same region but from two different provinces, namely Salerno and Caserta. These authors applied a discriminant analysis model based on four metabolites linked to milk processing (β-galactose, β-lactose, acetic acid, and glycerol) and, after cross-validation, 100% of samples were correctly classified. Similarly, Consonni and Cagliani [72] correctly differentiated 93.5% of Italian Parmigiano Reggiano PDO cheeses from foreign eastern European “Grana type” cheeses by applying a PLS-DA model on leucine, isoleucine, lactate, butanoate, and acetate. The same authors also discriminated Parmigiano Reggiano PDO cheeses based on their ripening times. However, no cross-validation was performed to test the developed model.

The simultaneous analysis of proteins, lipids, and other metabolite fractions by ^1^H HRMAS NMR offers great opportunities for cheese geographical origin authentications [73]. Moreover, NMR offers multiple advantages, such as simple sample preparation, multiple metabolite quantifications, high experimental reproducibility, and it is also non-destructive. Nevertheless, some limits should be considered, such as high cost for acquisition and maintenance, and higher limits of detection (typically, 10 to 100 times) when compared to gas chromatography-mass spectrometry [73,74]. Table 4 reports the advantages and limitations of nuclear magnetic resonance for cheese origin authentication.

### 2.5. Gas Chromatography-Based Fatty Acid Analysis

Recently, fatty acid analysis has been investigated for cheese origin authentication purposes. Fatty acids are lipid components formed by carboxylic acids with saturated or unsaturated aliphatic carbon chains [75]. Generally, fatty acid analysis is based on four sequential steps: extraction, derivatization, chromatographic separation, and detection [75]. The analytical reference method for fatty acid analysis is gas chromatography (GC), which is usually combined with flame ionization detectors [76]. For GC, analytes are vaporized in a heated chamber and transported by high-pressure inert gas (e.g., N^2^, He) through the stationary phase (i.e., column material), where selective interaction leads to compound separation [77]. Subsequently, based on their retention index, compounds are eluted into the hydrogen flame of the detector, creating an electrical signal [78].

Fatty acid analysis to authenticate milk origin has already been reported [79]. These authors were able to efficiently authenticate the geographical origin of milk as the combination of different feeding strategies, herd and farm management practices (leading to distinct feed fatty acid profiles), grazing, breeding, animals’ genetics, animals’ rumen microbiota, and difference in lactation days varied considerably according to geographic locations. In traditional cheese, milk fatty acid profiles are also impacted by manufacturing practices (e.g., use of *Cynara cardunculus* L. as a coagulating agent), cheese microbiota (see the section on DNA-based methods for cheese origin authentication), and ripening times [80,81]. For example, fatty acid profiles were used and correctly authenticated the producer origins of Serra da Estrela PDO cheeses from Portugal, even within a limited production area [76]. These authors used 12 fatty acids, namely caproic, caprylic, undecanoic, lauric, pentadecanoic, palmitic, palmitoleic, heptadecanoic, oleic, linoleic trans-isomer, heneicosanoic, and arachidonic acids, in a linear discriminant model to achieve an 88% correct classification rate after cross-validation. Higher classification rates (95% after cross-validation) were also reported by Margalho et al. [82] using 13 fatty acids for 11 artisanal cheeses produced in 5 major geographical regions in Brazil (sample size *n* = 402). Similarly, high classification percentages were also achieved by Danezis et al. [83], who analyzed 101 PDO and 11 non-PDO cheeses from Greece. These authors used pH, moisture, fat, NaCl, and linoleic acid contents to correctly discriminate the PDO from non-PDO cheeses (excluding similar hard PDO cheese) and achieved 100% correct classifications. Although high correct classification rates were obtained by Margalho et al. [82] and Danezis et al. [83] for cheese origin authentication, the analyzed cheeses greatly differed in terms of physical aspect (i.e., spread, soft, semi-hard, and hard), fat content, and ripening time. In recent studies dealing with origin authentication of similar traditional cheeses, the correct classification rates reported by Reis Lima et al. [76], Gatzias et al. [84], and Vatavali et al. [85] were, respectively, 88%, 88.2%, and 91.1%.

### 2.6. Gas Chromatography-Based Volatilome Analysis

Among the different strategies to authenticate cheese origin, volatile organic compound (VOC) analysis has gained attention as volatile compounds, which result from cheese microbiota metabolic activities, are an important component of cheese typicity. Indeed, through glycolysis, proteolysis, and lipolysis, cheese microbiota produce a wide range of VOCs. These include aldehydes, ketones, alcohols, esters, lactones, hydrocarbons, free fatty acids (*n* < 10 carbon atoms), sulfur compounds, and amines that provide the typical cheese aroma [86,87,88]. In this context, GC-MS is most commonly used to analyze cheese volatilomes (see Medina et al. [89] for technical details on cheese VOC analysis using GC-MS).

Using volatile fingerprinting, Pillonel et al. [90] discriminated country of origin for both PDO and non-PDO Emmental cheeses based on butan-2-one, 3-hydroxybutanone, butan-2-ol, and octene concentrations by principal component analysis (PCA). The concentrations of 21 other volatile compounds also showed at least one significant difference connected with their origin. Cheese origin discrimination relied on the fact that volatile profiles varied both qualitatively and quantitatively according to their country or region of origin [90]. For example, PDO Emmental from Switzerland was differentiated from Polish and French Emmental cheeses based on free-fatty acid qualitative composition (2-methyl butanoic acid for PDO Emmental cheese compared with 3-methylbutanoic acid in French and Polish Emmental ones) and relative abundance (such as nonanoic acid), alcohol presence/absence (3 methylbut-2-en-1-ol presence only in Swiss Emmental), as well as other aliphatic hydrocarbons, ketones, aldehydes, and esters. More recently, Pluta-Kubica et al. [91] also differentiated Emmental cheese origin based on their VOC profiles.

Similarly, Salzano et al. [37] used GC-MS to authenticate water buffalo mozzarella PDO cheese from the non-PDO versions. Both milk and cheese samples were analyzed, and differences were highlighted for both matrices using partial least squares discriminant analysis (PLS-DA). Variable importance in projection (VIP) analysis selected the 15 highest scored variables. Among them, talopyranose, 2,3-dihydroxypropyl icosanoate, sorbose, 4-phenyl glutamic acid, oxalic acid, and galactose were the most prevalent in typical PDO mozzarella, while tagatose, lactic acid dimer, ribitol, dodecyl thioglycolate, n-acetyl glucosamine, valine, and diethylene glycol were more abundant in non-PDO mozzarella. These authors concluded that the combination of multiple practices, such as forage from the same region, natural milk starters with both LAB and yeast instead of citric acid, and different packaging, all impacted the volatilome of the final product. These differences could thus explain what distinguished the water buffalo mozzarella produced according to PDO rules vs. those not following such guidelines (i.e., non-PDO mozzarella).

Another study authenticated Pecorino cheese origins, namely Pecorino Romano PDO, Pecorino Sardo PDO, and Pecorino di Farindola (certified by the Slow Food Foundation) [92]. The authors compared VOC fingerprints by a linear discriminant analysis (LDA) and PLS-DA model. The most influential variables in the LDA model were 2-methyl butyl isovalerate, butan-2-one, butyl butanoate, ethyl acetate, nonan-2-one, and propan-2-one, while in the PLS-DA model, VIP analysis identified 14 relevant compounds, namely, butan-2-one, pentan-2-ol, ethyl acetate, dicaprylyl ether, propanoic acid, 3-methylbutan-1-ol, propan-2-ol, ethyl decanoate, heptan-2-ol, butan-2-ol, butyl butanoate, pentan-2-one, ethanol, and 2-methylpropanoic acid. Only six compounds were common to the two tested models. Considering that both models yielded similar classification performances (total classification rate of 87.5% after cross-validation), differences in the most influential variables linked to origin authentication were related to the index score value applied in the VIP analysis. Indeed, while the authors used an index score of 1, a previous study by Salzano et al. [37] used a higher value (i.e., 1.5) to select variables of interest and obtain a high correct classification rate for water buffalo mozzarella. Noteworthy, Vatavali et al. [85] only classified 47.5% of Graviera cheese origins, and thus the discriminative power of VOC profiles for cheese authentication could vary considerably according to the considered cheese type.

GC-MS fingerprints can also be exploited to identify biomarkers connected with specific attributes of traditional products, such as animal feed requirements. Indeed, Caligiani et al. [93] validated a method to quantify cyclopropane fatty acids (e.g., dihydrosterculic acid) as biomarkers for cows fed with corn silage. PDO cheeses such as Parmigiano Reggiano, Fontina, Comté, and Gruyère do not permit silage to be used in cow feed, and thus the absence of cyclopropane fatty acids in such cheeses may confirm correct feeding management.

Another feed-associated fraction of cheese VOCs are terpenoids. Terpenoids are a highly diversified class of naturally occurring organic compounds or phytochemicals, also called isoprenoids. These are derived from isoprene units and produced by dicotyledon plants [88]. In Slovenian cheeses, VOC analysis discriminated cheeses into 4 clusters (average silhouette 0.764) according to their geographical origin and based on 9 monoterpenes, namely, α-pinene, camphene, α-phellandrene, β-pinene, 3-carene, 2-carene, limonene, tricyclene, and γ-terpinene [94]. In a similar way, Turri et al. [88] identified significant differences in 10 terpenes between pasture-producer of Historic Rebel cheese, and the results suggested that allo-ocimene, α-terpinolene, α-pinene, and δ-3-carene could be possible biomarkers to differentiate cheese origin. Overall, these studies highlighted that volatilome analysis can be an interesting tool for cheese origin authentication, although classification rate performances can vary among cheese varieties.

Another approach to directly analyze volatile compounds is the electronic nose (e-nose). This analytical technology, designed to mimic the human olfactory system [95], has gained interest in food authentication as it is highly correlated with consumer perception [96]. A typical e-nose comprises the sampling system, a set of non-selective sensors or mass spectrometer (MS), and a pattern-recognition system [97,98]. Nowadays, different sensors are used, such as metal-oxide semiconductors, conducting polymers, and piezoelectric crystal sensors [99]. In the case of cheese origin authentication, only a limited number of studies reported an e-nose strategy. For example, Pillonel et al. [90] obtained similar classification rates using a PCA model to evaluate Emmental cheese origins (90%). In the case of Pecorino cheeses, an e-nose and artificial neural network approach correctly classified 96.5% of Pecorino di Fossa PDO cheeses (*n* = 18) and Pecorino cheeses of other origins (*n* = 48) [100]. In conclusion, while different authors have reported e-noses to authenticate foods subjected to different frauds, including geographical origin [98,101,102], only a limited number of studies concerned cheese origin authentication. Table 5 reports the advantages and limitations of VOC analysis for cheese origin authentication.

Some important attributes that may impact the efficiency of this approach to discriminate cheese origin are provided. One important attribute to differentiate cheese is ripening time [106]. Typical cheeses are sold according to a minimum ripening time. However, according to the initial extent of ripening, significant changes may occur during shelf-life. Contamination during portioning, inadequate temperature usage during transport, light exposure, and storage conditions may impact VOC profiles. In this context, specific VOC fractions, such as terpenoids, combined with VOC and microbiota correlations, may overcome possible changes in VOC profiles at the retailing stage. However, to our best knowledge, no studies have evaluated the effect of retail on cheese origin authentication.

## 3. DNA-Based Methods for Cheese Origin Authentication

As previously mentioned, the metabolic activities of cheese microbiota play a crucial role in the development of cheese typicity. For geographical origin authentication, microbiota fingerprinting is therefore of high interest as traditional and artisanal cheeses are produced with a more diversified microbiota associated with the cheese-making process (e.g., use of raw milk, starter, brine, equipment and materials, and ripening rooms). Distinct differences in the composition of this complex microbiota, composed of Gram-positive and -negative bacteria, fungi, archaea, and viruses, could be used for cheese origin authentication.

The first study on cheese microbiota using high-throughput sequencing (HTS) was performed by Quigley et al. [107] on traditional cheeses. Since then, numerous studies have been published on many aspects linked to cheese quality and typicity. While cheese microbial diversity was traditionally investigated using culture-dependent methods, hence overlooking unculturable or subdominant species, nowadays, culture-independent methods (HTS) have unraveled this diversity and provided further means to connect microbiota composition to cheese quality and typicity, but also origin. This success is due to the availability of new sequencing platforms, bioinformatic pipelines, and a continuous decrease in cost. Among high-throughput sequencing, amplicon sequencing—or selective amplification of polymorphic genes across their hypervariable regions—is the most widely reported in the scientific literature [108]. In this context, the use of DNA metabarcoding (also known as metagenetics) to study cheese microbiota was proposed as a tool for cheese origin authentication [33].

To perform DNA metabarcoding, cheese samples are first homogenized, then total DNA is most frequently extracted using commercial kits, ad hoc protocols, or a combination thereof [109,110,111]. Hypervariable regions of taxonomically relevant genes (e.g., 16S rDNA for bacteria and archaea, ITS, 18S rDNA, 26S rDNA for fungi) are amplified by PCR reactions, while a second amplification step tags amplicons with specific DNA fragments—barcodes—and dedicated adapters for the final sequencing step using next-generation technologies (e.g., Illumina, Pacbio, Iontorrent, or Nanopore). For further details on sampling, library preparation, and sequencing platforms, the reviews by Hugerth et al. [112] and Tilocca et al. [113] are suggested.

Typically, 16S rDNA and ITS (internal transcribed spacer) markers, targeting bacteria and fungi, respectively, are employed to generate compositional data describing microbial taxa and their relative abundance in cheese microbial communities. After sequencing, two complementary but different ways can be used for amplicon clustering from quality-checked data, namely, operational taxonomic units (OTUs) and amplicon sequencing variants (ASVs) [114]. On one hand, pipelines such as QIIME and IMNGS build sequence clusters based on their similarity (usually using a similarity cutoff of 97%) to obtain OTUs [115,116]. On the other hand, ASVs are obtained with the DADA2 pipeline (also available in QIIME2) by inferring biological sequences in a sample, and discerning sequence variant differences down to a single nucleotide [117,118]. Subsequently, taxonomic assignment is performed using a specific classifier tool (BLAST, RDP, UCLUST, SortMeRNA) against various reference databases, such as Greengenes, SILVA, and UNITE [119,120]. Generally, clustered OTUs/ASVs are analyzed from the phylum to the genus level since they can be less precise at the species level [121]. Identified taxa can be divided into dominant, subdominant, and rare sequences, representing 40% to 90%, 1% to 0.01%, or 0.01% to 0.0001% of reads per sample, respectively [122].

Using these DNA methods to determine cheese microbiota composition, literature data have shown that the core is usually dominated by lactic acid bacteria (LAB) [123], including starter and non-starter lactic acid bacteria (NSLAB), while in the case of cheese with edible rinds, rinds are usually dominated by salt-tolerant fungi and bacteria from the *Actinomycetota*, *Bacillota,* and *Pseudomonadota* phyla. Cheese microorganisms can be deliberately inoculated into the milk or on the cheese surface as starter or secondary cultures, but may also originate from multiple reservoirs, including raw milk (i.e., for raw milk cheeses), brine or salt, or dairy environments (cheese-making equipment and ripening shelves). A distinctive case corresponds to raw milk cheeses as cheese microbiota can be differentiated according to the amount of starter cultures employed during cheese-making and the origin of the raw milk used, at the farm level [124]. Since milk quality depends on many factors (e.g., animal health status, breed, lactation stage, teat skin, hides, feces management, farm dimension, feeding system, season, farm staff hygiene, and management), recent longitudinal studies have connected some of the main characteristics with farm origin [125,126,127]. Indeed, for raw milk cheeses, cheese microbiota can be directly impacted by raw milk microbiota [107], thus, both microbiota can be used for cheese origin authentication. Generally, most protected land- and tradition-related labeled cheeses are produced from raw milk, and the complex microbiota encountered in raw milk directly influences the unique cheese sensorial properties appreciated by consumers. As an example, in the EU, among the 284 cheeses recognized for their typicity, over 180 are raw milk cheeses ([128] https://ec.europa.eu/info/food-farming-fisheries/food-safety-and-quality/certification/quality-labels/geographical-indications-register/ accessed on 3 October 2022). Besides raw milk, traditional tools/equipment and the dairy environment are also shaping cheese microbiota at both species and strain levels. In fact, some key species may originate from the dairy environment [129]. In addition, as reported by Bokulich and Mills [130] and Calasso et al. [131] for LAB, different strains of a given species can colonize the dairy environment, and thus, the cheese. More recently, Sun et al. [132] determined that in-house microbiota were essential in shaping Bethlehem (PA, USA), a Saint-Nectaire-type cheese produced without starters by traditional methods. Using 16S rRNA amplicon sequencing and SourceTracker (a bioinformatic tool based on Bayesian inference that estimates the proportion of different sources contributing to a designated microbial community) [133], these authors identified wooden vats—used for overnight ripening—as a major source of desirable LAB that shaped cheese microbiota from acidification to ripening. Similar findings were reported by Montel et al. [134], who highlighted that traditional cheeses have complex and rich microbiota influenced by traditional equipment. Indeed, traditional cheeses are produced using cheese-making practices that tend to increase microbial diversity via contact with diverse microorganisms originating from dairy equipment [135]. Moreover, differences in cheese-making technologies (e.g., use of natural milk or whey culture rather than commercial starters, use of rennet or clotting agents, curd cooking, draining process, salting) and farm practices (e.g., type of housing, silage, grassland) between different production areas can also shape the cheese microbial community [136]. Considering these factors, we can question whether traditional cheeses can be differentiated based on their origin and if the main factors that affect cheese microbiota complexity and diversity play a significant role.

### 3.1. Main Factors Affecting Cheese Microbial Diversity

In this context, in a recent study by Kamimura et al. [137], the microbiota of 578 traditional Brazilian cheeses were analyzed with amplicon sequencing. Bacterial communities were distinctly clustered with PCA by cheese type and regional origins while, at the genus level, hierarchical cluster analysis separated production regions. These authors were thus able to identify specific origin-related microbiota. The core microbiota of Brazilian traditional cheeses displayed different relative abundances and oligotypes (i.e., closely related but distinct bacterial taxa) of LAB belonging to the *Enterococcus*, *Lactococcus*, *Streptococcus*, *Leuconostoc*, and *Lactobacillus sensu lato* genera, as well as other taxa belonging to the *Enterobacteriaceae* family and *Staphylococcus* genus. Within the same regional area, microbiota analysis differentiated the origin—Cerrado, Araxà, Canastra, Campos des Vertenes, and Serro—of traditional cheeses produced with a similar natural whey starter and ripening period (17 and 22 days). These findings were in agreement with those of another study that analyzed 97 samples of Minas artisanal cheese from 6 different producers located in the same region [138]. Starter cultures, consisting of *Streptococcus*, *Lactococcus*, and *Lactobacillus sensu lato* spp., constituted the core microbiota in all farms. However, significant differences in family- and genus-level bacterial community relative abundances were observed between the studied farms due to environmental factors such as geographical location. Even when dominant genera may be inferred to the natural whey cultures used, meta-analysis from amplicon sequencing data of traditional artisanal cheeses from Italy, Belgium, and Kalmykia indicated differences in bacterial structures between cheeses produced in different geographic areas (unweighted PCoA cluster MANOVA, *p* < 0.001 [139]). Those produced using natural milk cultures showed improved acidification without an effect on the typical cheese microbiota. Indeed, at the end of the ripening period, cheese origins clustered according to producer facilities (PCoA on Bray–Curtis) [140].

Another study was performed by Zago et al. [136]. In this case, 118 Grana Padano samples were analyzed after 7–8 months of ripening and a common core microbiota composed of *Lactobacillus-*, *Lactococcus-*, *Lacticaseibacillus-*, *Limosilactobacillus-*, and *Streptococcus*-dominant genera was observed. More precisely, differences in bacterial abundance, richness, and evenness were found for dominant and sub-dominant groups according to production region, a result also confirmed by PERMANOVA beta-diversity analysis. The authors also identified specific species that could be linked to several production areas; however, no species biomarkers were identified regardless of production area and non-metric multidimensional scaling did not show any clear clustering profile.

Some cheeses are produced in very small geographical zones by a limited number of producers. This is the case for Plaisentif and Historic Rebel cheeses from the mountainous regions in Italy, that are only produced during specific seasons (violet blooming season and grazing season) by 14 and 12 producers, respectively [88,122]. Both are raw milk cheeses produced without starter adjunction. Bacterial amplicon sequencing analysis (16S rDNA V4 region) for Plaisentif cheese identified dominant genera and, more importantly, differences in bacterial community profiles between producers thus detected fraudulent starter additions in some cheeses [122].

Based on a similar analysis for Historical Rebel cheese, the core microbiota was composed of 5 different genera—*Streptococcus*, *Lactobacillus*, *Lactococcus*, *Leuconostoc,* and *Pediococcus*—with *Streptococcus* relative abundances ranging from 60% to 85% [88]. Richness and other alpha-diversity parameters differed among producers as well as in multivariate analysis (PCoA on unweighted Unifrac), and based on the observed significant differences, pasture area could be linked to the different Historic Rebel cheese producers.

### 3.2. Climatic and Environmental Condition

Another factor that can impact microbial communities of traditional cheeses are the climatic and dairy environment conditions that are directly associated with geographical origins. This was observed for traditional Chinese Rushan cheese produced using *Chaenomeles sinensis* boiled extract as a clotting agent in three different regions. Even if the same UHT milk and production equipment were used, geographical origins significantly impacted the relative abundance of 12 dominant genera, namely *Lactobacillus*, *Acinetobacter*, *Acetobacter*, *Lactococcus*, *Enterobacter*, *Moraxella*, *Enterococcus*, *Streptococcus*, *Kocuria*, *Staphylococcus*, *Chryseobacterium*, and *Exiguobacterium* [141], and is likely related to specific house microbiota and open-air drying. This result was also confirmed by PCoA clusters and Anosim analysis.

As typical cheese microbiota can be in part acquired from the specific raw materials used, traditional tools, environmental and production conditions, cheese-making process, and geographical area, a comparison between traditional and industrial cheeses may provide additional information to authenticate cheese origin. Noteworthy, some authors have reported that commercial starters, inoculated at ~10^6^ CFU/mL, prevent resident microbiota from developing, especially during ripening [124,142]. Overall, milk pasteurization, use of similar commercial starters, similar industrial equipment, and standardized recipes for cheese production are crucial factors that decrease cheese microbial complexity and biodiversity and lead to highly standardized productions. These directly deplete the unicity of the matrix, and thus lower variance is detected (Figure 1). This hypothesis is in accordance with the study by Kamilari et al. [143], in which a significant decrease in bacterial diversity was observed in industrially produced Haloumi cheeses vs. artisanal products. However, the microbial diversity observed for artisanal Haloumi cheeses could not link them to their producers’ geographical origins. In another study, aiming to authenticate cheese origin at the producer level, no distinction could again be made [144].

### 3.3. Cheese Ripening

Cheese ripening is another factor that affects microbial community diversity and cheese typicity. Ripening can be considered as a selection process that leads to cheese microbial composition changes. According to Gobetti et al. [123], intentionally added microorganisms used in cheese-making include primary starters (natural milk culture, natural whey culture, or lyophilized commercial starters), secondary or adjunct LAB starters, and milk autochthonous microbiota (NSLAB and others). These are the main ripening agents in intermediate to long ripening times, which mainly explain the observed diversity and typicity of the produced cheeses. The relationship between primary starters and NSLAB during maturation is well-known and involves a progressive reduction of the former in favor of the latter. The role of NSLAB is crucial in maturation and the development of the typical characteristics of traditional cheese. In this case, useful insights can be gained by comparing the genomic features of primary starters for the presence of genes involved in the metabolic pathways important for cheese maturation. While primary starters have important genetic features for the utilization of lactose—mainly connected with their acidification ability—NSLAB possess many more genes coding for peptidases, peptide transporters within cells, and amino acid catabolism, that can represent an advantage during cheese maturation [145]. Moreover, compared to primary starters, NSLAB tend to adapt better to the hostile conditions of the cheese ripening environment, such as temperature, salt content, pH, and redox potential. In fact, NSLAB can adopt alternative metabolic pathways to produce energy from unconventional sources while resisting acid conditions. Therefore, NSLAB present in raw milk at a sufficient inoculum to colonize ripened raw milk cheeses, or acquired from the house microbiota, could be an indicator of geographical origin at the producer level. Beyond NSLAB, other microorganisms belonging to various groups can influence cheese ripening. This is the case of fungal communities in many traditional cheeses, such as Queijo de Azeitão in Portugal [146], Tomme d’Orchies in France [110], and Robiola di Roccaverano in Italy [147]. Indeed, fungal communities are well-known for their decisive role in flavor and texture of white and blue-veined mold-ripened cheeses due to lipolytic, proteolytic, and glycolytic activities, leading to high production of aromatic ketones and alcohols [134,148,149]. Generally, fungal species such as *Penicillium camemberti*, *Penicillium roqueforti*, *Debaryomyces hansenii*, *Kluyveromyces marxianus*, *Candida catenulata*, *Galactomyces geotrichum*, and *Mucor lanceolatus* are either deliberately added as technological adjunct cultures or present in the production environment [149,150,151,152]. Nevertheless, in traditional cheeses, fungal communities were reported to be more diverse than the used starter but, at the same time, not connected with geographical origin [109,146]. Table 6 reports advantages and limitations of amplicon sequencing for cheese origin authentication.

The mentioned studies showed that in traditional cheeses, the combination of artisanal cheese-making, specific raw materials, and characteristic environmental conditions shape microbial community diversity according to geographical origin. Most analyses conducted using 16S rDNA amplicon sequencing discriminated cheese origin, although taxonomic classification was still limited to genus/family-level descriptions and only a few cheese types per study were considered. To further assess the unicity of typical cheeses against food fraud, more in-depth studies, including meta-analyses on all available cheese data and increased depth of microbial population descriptions (e.g., metagenomics), are of interest.

## 4. Conclusions and Perspectives

Geographical origin authentication is an important safeguard for food quality and safety but also from an economical point of view as it enables consumer protection and provides technical support to enforce national and international legislations [49]. Applying elemental and isotopic characterization, volatilome or microbiota analysis to typical products can protect them from food fraud and improve registration processes and marketing decision-making [161]. Nevertheless, our knowledge on authentication methods is far from complete. Indeed, most studies and methodologies employed to authenticate cheese origins only provide qualitative answers (e.g., does the method discriminate origin?) and often lack quantitative assessment (to what extent can different cheese origins be discriminated from each other?). This is probably connected to the complexity of commonly used multivariate models such as PLS, PLS-DA, and LDA, and the use of specific algorithms (e.g., Random Forest, VIP) for variable selection and the need for internal or interlaboratory external and cross-validations.

Chemometrics approaches differ among analytical technologies. In this case, infrared methods—especially NIR spectroscopy—have consistently been applied together with chemometrics analysis to achieve good classification rates after cross-validation, but the number of samples employed for authentication purposes was often limited. Stable isotope ratios combined with trace element analysis have been shown to be the most accurate methods to authenticate cheese origin, since available studies reported consistent statistical analysis and modeling with high correct classification rates after cross-validation. Nevertheless, the actual discriminative power of the method for closely distant cheese producers remains unknown. Moreover, the impact of animal feeding and the high cost per sample must be considered. Considering that many production disciplinaries declare a minimum amount of local forage in animal feed, one possible strategy to increase cheese typicity and improve authentication using stable isotopic and elemental metabolomics would be to increase the use of local feed or use grazing.

Regarding DNA-based methods, amplicon sequencing can discriminate cheese geographical origin. However, some important considerations to assess traditional cheese origin are related to metabarcoding approach limitations. Indeed, microbial species and strains that originate from the dairy environment and that characterize traditional product origins could be key biomarkers for cheese authentication. In this context, shotgun metagenomics offers multiple advantages in comparison with amplicon metabarcoding, since no amplification step is required and all the genetic material in the sample is used for genome reconstruction to reach a deeper taxonomic assignation, potentially to the strain level [162,163]. For example, StrainPhlAn uses unique gene family markers and sample-specific consensus sequences to infer strain-level genotypes from different environments [164]. Considering that cheese is formed by microorganisms of different domains, appropriate sequencing depths would directly provide insight on bacteria, fungi, and viruses in the same analysis. These tools could thus be applied to milk, starters, and typical cheeses to obtain accurate DNA-based fingerprints to efficiently authenticate product origin [164]. This approach may help explore new traceability systems based on crucial components of fermented products, such as their virome [165]. Moreover, shotgun metagenomic analyses of gene richness, as a possible indicator of microbial community adaptation to different stress conditions, would be useful to reconstruct metabolic pathways connected with specific cheese traits, such as volatile compound production and metabolites that characterize typical cheeses. In this sense, integrated systems biology, combining metabolomics and metagenomics, could improve our knowledge on this subject, as only a few studies to date have combined both techniques [88,156,166]. While microbiota-based studies have compared typical and non-typical products, some omics approaches reported differences between typical and industrial products, thus making it difficult to clearly determine the factors that characterize typical food products. Artificial intelligence approaches, such as deep learning and machine learning, should be taken into consideration to improve classification rates and better-differentiate authentic and fraudulent products [167]. Overall, further research focused on comparing how well DNA-based analyses perform in comparison to the actual reference analyses (i.e., isotope fingerprinting and trace element analysis) used to authenticate cheese origin is needed. A combined approach, using isotope fingerprinting or trace element analysis and metagenomics, to obtain the highest discriminative power for cheese geographical origin authentication could also be of interest.

## Figures and Tables

**Figure 1 foods-11-03379-f001:**
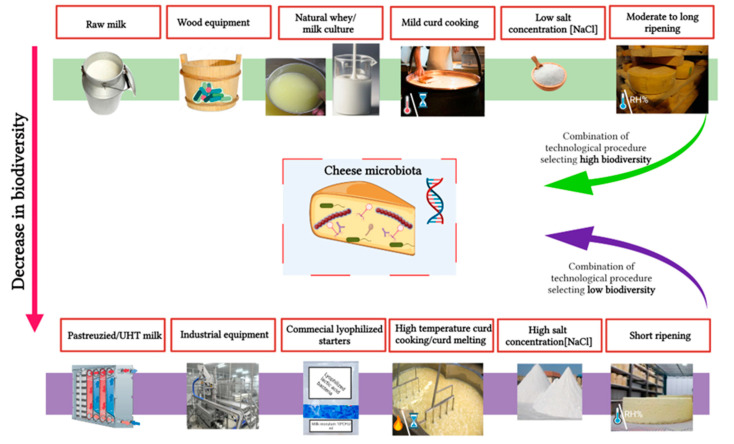
Technological factors affecting cheese microbiota biodiversity. Green and purple lines show combinations of technological factors during cheese-making that increase or decrease this biodiversity.

**Table 1 foods-11-03379-t001:** Advantages and limitations of stable isotope ratio mass spectrometry for cheese origin authentication.

Advantages	Reference	Limitations	Reference
Isotope ratio such as ^18^O, ^2^H, and ^34^S have a predictable and reproducible response toward geographical origin	[38]	Unclear effect of cheese-making on stable isotope ratio	[34,43]
Elevate correct classification rate	[42]	High operating cost	[47]
Consistent accuracy in origin authentication	[34,41,42]	^13^C and ^15^N are highly affected by animal feed	[41]

**Table 2 foods-11-03379-t002:** Advantages and limitations of inductively coupled plasma-mass spectrometry for cheese origin authentication.

Advantages	Reference	Limitations	Reference
Elemental composition is mainly affected by geological and pedological traits	[49]	Animal feed and mineral supplementation affect elements’ composition	[50]
Low operation costs with good analytical performance	[47]	Some elements such as Cu^2+^ and Zn^2+^ are highly affected by dairy equipment	[52]
Fast and multi-element analysis	[44]	Requires careful sample preparation	[39]

**Table 3 foods-11-03379-t003:** Advantages and limitations of infrared spectroscopy for cheese origin authentication.

Advantages	Reference	Limitations	Reference
May identify peculiar spectral differences connected with cheese origin	[57]	Requires large sample size for calibration	[64]
Fast, solvent-free, automatic, non-destructive, non-invasive, inexpensive, and can be used as a multiparameter analysis	[49]	Fingerprints may not detect less concentrated molecules connected with geographical origin	[57]
Minimal sample preparation	[58]	Lack of studies on the analysis and quantification the main sources of absorbance variation at each wavelength for cheese	[65]

**Table 4 foods-11-03379-t004:** Advantages and limitations of nuclear magnetic resonance for cheese origin authentication.

Advantages	Reference	Limitations	Reference
Allows to obtain detailed information on cheese metabolites	[44]	Extremely high cost of acquisition	[73]
May identify biomarkers related to geographical origin	[73]	High detection limits	[74]
Minimal sample preparation	[49]	Complex calibration	[73]

**Table 5 foods-11-03379-t005:** Advantages and limitations of volatilome analysis for cheese origin authentication.

Advantages	Reference	Limitations	Reference
High resolution, short separation time, high sensitivity, and low cost	[32]	High dependency on ripening time	[103,104]
Measures correlated with cheese quality	[87]	Highly impacted by extraction methods	[89]
Fingerprint profile with discrete correct classification rate	[92,105]	High variability of correct classification rate	[37,85,92]

**Table 6 foods-11-03379-t006:** Advantages and limitations of HTS amplicon sequencing for cheese origin authentication.

Advantages	Reference	Limitations	Reference
Time- and cost-effective processing of large sample numbers	[153]	Analyses could be biased by sample processing, DNA extraction methods, and equimolar library preparation	[154]
Consolidated pipeline for data analysis	[155]	PCR amplification steps include errors, e.g., PCR specificity and variation of 16S rRNA copy number per genome	[108]
Identification of taxonomic groups associated with typical flavor and cheese-making technology	[156]	Under- or over-estimation of microbial community diversity	[112,155]
Allows improvement of cheese-making to ensure safety while preserving typicity	[137]	Lack of absolute abundance	[157,158]
Evaluation of core microbiome describing facility-associated microbial groups	[130,131]	Limited and uneven taxonomic resolutions	[159,160]
May pinpoint new biotypes	[143]	DNA amplicon sequencing typically does not discriminate between live and dead microorganisms (except if DNA stains such as propidium monoazide are used)	[87]

## Data Availability

Not applicable.

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
