# Peer review of "Authenticity and Typicity of Traditional Cheeses: A Review on Geographical Origin Authentication Methods"

_foods, 2022, doi:10.3390/foods11213379_

Round 1

Reviewer 1 Report

The present review study aims at presenting the collection of methods, both physical and chemical, generally employed in the endeavor of authenticating cheese based on the geographical origin.

Overall, the paper is well written and structured throughout. The general introduction of the context is clearly identified in the Introduction section, and the main points are covered and recapitulated towards the end.

Given that the article deals in detail with the studies at the interface between analytical fingerprinting of cheese (IRMS, ICP, IR spectroscopy, NMR, GC, and DNA-based) and chemometrics, I find it to be focused and critical; at the same time, it aligns with the need to be relevant to a broad audience. As the main focus of my feedback is the manner of conveying the message, I would like to note here the clear communication of the topic; language and phrasing are clear and unambiguous.

With regard to the presentation, the authors introduced and defined new terms, and used them consistently; in addition, all techniques and methods were thoroughly explained. I believe the study discusses the literature critically and identifies methodological problems; the research gaps are pinpointed by the use of limitations (Tables 1-6), adequately presented and discussed throughout.

Taken together, accompanied by detailed steps for data curation towards cheese geographical origin fingerprinting and authenticity purposes, the topic of the paper is relevant and of interest for the readers of the journal.

Author Response

Dear reviewer,

We have read your comments about our work and are very thankful.

Please see the attachment for the spelling corrections. 

Best regards,

The authors

Reviewer 2 Report

The review is very comprehensive and informative. Great job. Only a few mistypes need to be changed (see attachment). 

The main question addressed by the research is focused on the recent methods applied for the authentication of cheeses. Accordingly, the authors presented various chemical and physical methods such as stable isotope ratio mass spectrometry, inductive coupled plasma mass spectrometry, infrared spectroscopy, nuclear magnetic resonance, and volatile analysis, as well as, the advantages and limitations of these methods. Authors referred to DNA-based methods too. The manuscript is relevant and interesting. It describes many up-to-date methods recently used to try to distinguish various kinds of cheese according to their origin, way of production, and percentage of precision. The topic is original covering comprehensive up-to-date references that cover this very actual topic. The manuscript adds to the subject area a high-quality review that describes methods for cheese authentication and problems of cheese authentication. The paper is well-written and logically structured, easy to follow, and clear to read. Conclusions are consistent with the evidence and arguments presented emphasizing the problem of food (cheese) frauding and how to prevent it. They suggest a combined approach according to the novels in the available methods.

Author Response

Dear reviewer,

We have read your comments about our work and are very thankful.

All your suggestions have been taken into account in the revised manuscript.

Best regards, 

The authors
